# Stranding of Mesopelagic Fishes in the Canary Islands

**DOI:** 10.3390/ani12243465

**Published:** 2022-12-08

**Authors:** Airam N. Sarmiento-Lezcano, María Couret, Antoni Lombarte, María Pilar Olivar, José María Landeira, Santiago Hernández-León, Víctor M. Tuset

**Affiliations:** 1Instituto de Oceanografía y Cambio Global, IOCAG, Palmas de Gran Canaria, Unidad Asociada ULPGC-CSIC, Campus de Taliarte, Universidad de Las, 35214 Telde, Spain; 2Institut de Ciències del MarCSIC, Passeig Marítim 37–49, 08003 Barcelona, Spain

**Keywords:** myctophids, central-eastern Atlantic, stranding, otoliths, remote sensing

## Abstract

**Simple Summary:**

We investigated the causative mechanism of the first mesopelagic fish strandings along the southeast shore of Gran Canaria Island (Canary Islands) during June 2021. We examined remote sensor data (current velocity, Trade Winds, and the presence of upwelling filaments and eddies near the island) to determine the reasons for the strandings. The biological data collected was appropriate for external morphological identification and otolith analysis. In summary, the stranding of mesopelagic fishes was dominated mainly by *Diaphus dumerilli*, although the otolith analysis revealed the presence of other Myctophidae species. Stranding events are common and appear to be related to mesoscale oceanographic structures. A monitoring program of coastal sites in combination with satellite-derived environmental data will help to gather robust information for modeling when and where such events might occur and what species are associated with these processes.

**Abstract:**

Most mesopelagic fishes perform large diel vertical migrations from the deep-sea zone to the surface. Although there is a trade-off between a higher food availability at the upper layers and an energy cost and predation risk, incursion towards the surface also implies a transport by currents, where the fish are exposed to a stranding risk on the coast. Here, we reported the first documented stranding of mesopelagic fishes along the southeast shore of Gran Canaria Island. Our study hypothesized that (1) the influence of the Canary Current, (2) the dominant incidence of the Trade Winds during summer, and (3) the presence of an upwelling filament coupled with an anticyclonic eddy south of Gran Canaria Island were the causative mechanisms of the strandings. *Diaphus dumerilii* (Myctophidae family) was the main species found as observed from an external morphological analysis using traditional taxonomy. The otolith contour analysis suggested the presence of other *Diaphus* spp. and *Lobianchia dofleini*. Nevertheless, the otolith morphological features described in the literature suggested that all the specimens were actually *D. dumerelii*. Errors in the identification were mainly due to the high intraspecific variability found in the otolith morphology. Even so, two patterns of oval and elliptic shapes were described with significant differences in its morphometry.

## 1. Introduction

The mesopelagic, or twilight zone (water masses between 200 and 1000 m depth, Robinson et al. [1]), is considered as the oceanic region with the highest micronekton biomass, holding ~1000 million tons of mesopelagic fishes [2,3,4,5,6]. These are highly diverse and abundant and conform to an important community of deep-sea ecosystems as well as, at the intermediate trophic levels, top predators, a critical component of food webs [7,8]. Mesopelagic fishes comprise 30 families [9]. Myctophidae (lanternfishes) are one of the most diverse deep-sea groups (36 genera and 260 species, [10,11]). Moreover, this group is one of the most abundant marine vertebrates collected globally in mesopelagic waters [12]. Most mesopelagic fishes perform large diel vertical migrations (DVMs, [13]) from the deep-sea zone, where they remain during the day, to the shallower layers where they feed at night [14,15,16]. Thereby, they transport the ingested carbon in the upper productive layers to the deep waters [17,18,19], which is in turn of significance to the biogeochemistry of the ocean [3,20,21,22]

Stranded mesopelagic fishes have been recorded in Mediterranean waters along the Sicilian coast of the Strait of Messina (central Mediterranean) [23,24,25,26], providing important biological material for taxonomical studies since the eighteenth century [23,24,27,28]. Several studies of stranded species were carried out to relate the *sagittae* otolith size with the fish body size [24,27], the abundance of stranded organisms, and the causes of the stranding [27,29,30]. The upwelling of deep-water masses drags deep-sea organisms from their normal habitats. Then, tidal currents and favorable wind conditions are the main mechanisms of mesopelagic fish stranding in the straits [28]. Battaglia et al. [27,30] also reported that the phase of the moon promotes variability in both current strength and irradiance, thus affecting the vertical distribution of fishes.

In the Canary Islands, the distribution and DVM of epi- and mesopelagic species of fishes [31,32,33,34], decapods [32,33,35,36,37], cephalopods [32,33,34,38], and euphausiids [32,39] are well known. Recent studies around this archipelago have focused on the reproduction and growth of one the most abundant mesopelagic fish (*Notoscopelus resplendens*) [40,41], as well as the migratory pattern, vertical distribution, and diet of mesopelagic cephalopods (*Abralia veranyi* and *Abraliopsis morisii*) [42]. Furthermore, Ublein and Bordes [43] argued that topography also influenced the species distribution around these volcanic islands. Despite growing interest on the mesopelagic community in this area, strandings of mesopelagic fishes along the coast of the Canary Islands have never been documented before.

The aim of the present study was to report the first stranding event of mesopelagic fishes in the Canary Islands (east coast of Gran Canaria Island) and to provide information about the oceanographic conditions involved in the strandings. Specifically, some specimens were identified using taxonomic keys, although most of them were identified from the *sagittae* otolith shape. Additionally, otolith measurements were also taken to obtain the relationships among them. Finally, the remote sensing data of temperature, dissolved oxygen, current fields, wind, and net primary production were examined to explain the causative mechanisms of these strandings around the oceanic islands.

## 2. Materials and Methods

### 2.1. Collecting Samples

On 2 June 2021, myctophid fishes were found stranded along the “El Inglés” beach on the southern coast of Gran Canaria Island, Canary Islands (Figure 1). This beach corresponds to the eastern boundary of the Maspalomas dune field. Specimens (n = 432) were collected by hand from the sand approximately every 50 m along the coastline from north to south and were immediately stored at −20 °C. The time of exposure to the sun, beach cleaning services, and scavenger animals affected specimen condition. Consequently, only a reduced number of individuals (n = 22) were used for the taxonomic identification based on the number and position of photophores using traditional guides [44,45,46]. The remaining specimens were identified from the previously extracted *sagittae* otolith. The standard length (SL, in mm) was measured for each specimen.

### 2.2. Classification of Sagittae Otoliths

We used the AFORO website (http://isis.cmima.csic.es/aforo/ (accessed on 12 August 2022) [47]) for the identification of species using the otolith contour (n = 170). This website offers a wide-open online catalogue of otolith images (470) of myctophids (106 species) from the Mediterranean Sea and Atlantic Ocean. The online classifier is based on a contour analysis using wavelets [48,49] and the non-parametric k-means algorithm [47]. This database classified most of the similar species based on their otolith morphology, providing nine possible species. Considering the bias in using this database due to the number of images per species, we established the following criteria for the selection of species: (1) closest species based on otolith morphology, (2) number of times that a species was repeated among the nine possible solutions, and (3) the order of species appearance. Most specimens were assigned to *Diaphus* spp. (see results). Thus, we tested the final assignation with the morphological description provided by Schwarzhans [50] for this taxon. Otolith length (OL, in mm) and width (OW, in mm) were measured for each specimen as well as the aspect ratio (AR = OW/OL) and otolith relative length (OR = (100 × (OL/SL)) [51,52]. This protocol was also applied for the otoliths of specimens that were identified taxonomically.

The normality and homogeneity of variances of OL, OW, AR, and OR were checked for each morphotype using the Shapiro–Wilk test and the Bartlett test, respectively. The average value of both variables was compared using the Mann–Whitney U-test (non-parametric test) or Student’s *t*-test (parametric test). Analyses were performed using the package ggstatplot in R [52].

### 2.3. Oceanographic Characterization

We studied daily time-series data (from 1 February 2020 to 15 January 2022) of currents and winds at the surface (0.5 m depth) to examine the causative mechanisms of the strandings. The components of current (eastward and northward) and wind velocity in front of the sampled area (Station 1) were analyzed from the surface to a 1000 m depth. We also analyzed the sea surface temperature (SST, °C) and chlorophyll-a concentration (mg·m^−3^) from remote sensor data. We downloaded data (currents, wind and SST) from Copernicus Marine Environment Monitoring Service (http://marine.copernicus.eu/ (accessed on 30 August 2021)) with a spatial resolution of 9 × 9 km. Chlorophyll-a was obtained from NASA’s OceanColorWeb (https://oceancolor.gsfc.nasa.gov/ (accessed on 14 November 2022)) website. This had a spatial resolution of 4 × 4 km. We used the programming language R [52] to analyze and represent the results. The sampling map was generated using the geographic information system QGIS (V.3.22.3) [53].

## 3. Results

### 3.1. Taxonomic Classification and Otolith Shape

All the individuals with visible photophores were taxonomically identified as *Diaphus dumerilii*. Their otoliths presented a high variability in shape with few and strong ventral denticles and a round posterior region with a wide and variable postdorsal depression, which was slightly curved and moderately thick (Figure 2). The *rostrum* was slightly longer than *antirostrum*, showing variable shape and a wide *excisura ostii* with a shallow or absent notch. Two otoliths presented different morphologies to the patterns noted above, and they were considered rare (Figure 2). Overall, the morphological features displayed a high intraspecific variability in terms of the contour shape. Thus, the AFORO classifier assigned 62.5% of the otoliths to *D. dumerilii*, 33.3% to *D. problematicus*, and 4.2% to *D. fragilis*. When the classifier was used with the remaining otoliths (without being taxonomically identified), most individuals were identified as *D. dumerilii* (74%) and *D. problematicus* (11.3%) and to a lesser extent as *D. adenomus* (6.7%) *D. fragilis* (5.3%), *D. vanhoeffeni* (1.3%), *D. malayanus* (0.7%), and *Lobianchia dofleini* (0.7%) (Figure 3). However, taking into account the otolithic morphological features of these species, especially the curvature of the inner and outer face, we reassigned all the specimens as *D. dumerilii* except for the rare otoliths (Figure 2).

Two otolith contour outlines were differentiated in *D. dumerilii* depending on their dorsal and pre-dorsal rim: morpho-type 1 was classified when the dorsal rim was more developed (in the middle or closer to the anterior region) and the predorsal rim was strongly depressed, and morpho-type 2 was classified when the dorsal and predorsal rims were less developed and depressed in comparison to morpho-type 1 (Figure 2). Morpho-type 2 showed significantly higher values for *OL* (Mann–Whitney U-test = 2100, *p* < 0.001), *OW* (*t*-test = −2.76, *p* = 0.006), *AR* (*t*-test = −4.95, *p* < 0.001), and *OR* (*t*-test = −3.62, *p* < 0.001) than morpho-type 1 (Figure 4). This variability was not linked to differences in fish size between both morpho-types (Mann–Whitney U-test = 3144, *p* = 0.1456).

### 3.2. Oceanographic Conditions

An anticyclonic eddy was observed south of Gran Canaria, which recirculated offshore water towards the island coast during the previous days (26 May 2021; Figure 5A,C) and during the same day (2 June 2021; Figure 5B,D) of the stranding event. In addition, a filament of upwelling transporting colder water and chlorophyll from the African coast ended in the anticyclonic eddy shed by Gran Canaria Island (Figure 5C,D).

The total component of the current velocity and wind values (Figure 6) were higher near to the island (Station 1 and 2) than in the open ocean than the effect of the upwelling filament and the anticyclonic eddy (Figure 5B,D) during the previous month. Both the current and wind speed values were also high near to the island during May 2021. The direction of the current was almost the opposite of the effect of the convergence between the filament and the eddy south of the island (near the stranding site) (Figure 5).

## 4. Discussion

Our study was the first documented evidence of stranded mesopelagic fish in the Canary Islands. The stranded organisms were mostly *Diaphus dumerilii* (Myctophidae), an abundant species in this region [33,34,43]. The stranding of mesopelagic fishes is poorly documented worldwide, except for the Sicilian coast in the Strait of Messina in the Mediterranean Sea [23,26,54,55,56,57,58]. In this region, 32 species of the families Gonostomatidae, Microstomatidae, Myctophidae, Paralepididae, Phosichthyidae, Sternoptychidae, and Stomiidae were collected between 2008–2016 [57]. In the Canary Island waters, mesopelagic species consisting of ten species of Gonostomatidae, fifty-two Myctophidae, four Phosichthydae, nine Sternoptychidae, and fifty-two Stomidae were collected during oceanographic cruises [33,34,43], showing a high diversity in the area. The family Myctophidae was the most numerous taxon around the Canary Islands, and species such as *Diaphus holti*, *D. dumerilii*, and *D. metopoclampus* [27,57,58] showed the highest abundances. This genus contributes the highest number of species within the Myctophidae, being represented by 78 species in the world’s oceans, and 21 species are found in the equatorial and tropical Atlantic [46].

The identification of stranded fishes in the Strait of Messina was carried out according to the position of photophores [45]. Here, we identified *D. dumerilii* from the otolith contour. However, our identification was moderately successful due to the similarity of otoliths among the *Diaphus* species, especially in *D. fragilis* and *D. problematicus*. The main problems for identification were related to the short rostrum and elliptic shape of their otoliths in comparison with other species [50,59]. Furthermore, Schwarzhans [50] already described similarities in the morphological pattern of *D. fragilis* otoliths (including to *D. fragilis* and *D. problematicus*, among other species) and *D. garmani* groups (in which *D. dumerilii* is included). However, the thickness and curvature of the inner and outer faces allowed the reclassification of the specimens of *D. dumerilii* because the faces were more concave in *D. fragilis* and flattened in *D. problematicus* [50]. Thus, our review of otolith face concavity was of importance for elucidating the misclassifications, even the ones with an undulant dorsal rim, which share a morphological similarity with *D. pedemontanus* [60], a fossil species. In any case, the high variability in morphometry and the shape of the otoliths of *D. dumerilii* revealed intraspecific differences in growth. It is well documented that this variability is related to sexual dimorphism, feeding, or swimming ability, as was described in other fish species (e.g., in [59,61,62]).

The stranding of these species (*Diaphus* spp.) may be due to how they swim. According to the body and *sagittae* otolith shape [59], two morphotypes were determined (Diaphus-deep: *D. brachycephalus*, *D. vanhoeffeni*, *D.* holti, *D. mollis*, and *D. rafinesqui*; Diaphus-slender: *D. dumerilii*, *D. fragilis*, *D. metopoclampus,* and *D. problematicus*). These authors described that the characteristics of the Diaphus-deep morphotype improve the acceleration or explosive swimming of these fish even though they have a more robust body [62]. However, the Diaphus-slender morphotype has a more streamlined shape that improves their prolonged swimming ability [62]. In addition, according to the body shape [63] two additional morphotypes were described for *Diaphus* spp. (fusiform: *D. dumerilii*, *D. vanhoeffeni,* and *L. dofleini*; elongatum: *D. fragilis*). In our study, the species were classified as either the fusiform or Diaphus-slender morphotype, which facilitated a prolonged swimming that was not enough to overcome the strong currents in the area.

The analysis of the oceanographic scenario south of Gran Canaria Island showed the presence of an anticyclonic eddy and an upwelling filament before and during the period of stranding. Mesoscale structures leeward of the islands were described long ago [64,65,66]. Anticyclonic eddies act as retention zones for phytoplankton [64,67,68,69,70,71], zooplankton [72,73,74,75], invertebrate larvae [76], and fish larvae [72,77,78], affecting their distribution [77,79,80]. The upwelling of cold deep waters in the African coastal zone depends on the intensity of the Trade Winds. North of 25° N, the upwelling intensifies during summer [69,71,80,81] as the Trade Winds increase during this season. Upwelling filaments are generated by the interaction of eddies and the coast mostly during the Trade Wind season, and they also promote the offshore transport of phytoplankton [64,67,68,69,70,71], zooplankton [82,83], invertebrate larvae, and ichthyoplankton [72,77,78,79,84] from the African neritic zone, eventually reaching the Canary archipelago [68,69,70,77,83,84,85,86]. The filaments are relatively shallow structures less than 100 m deep, which can extend over hundreds of kilometers into the ocean [83]. Therefore, transport in the upwelling filament and retention in mesoscale structures such as the anticyclonic eddy were suggested to promote the transport of mesopelagic fishes to the coastal zone where these organisms were collected for this study. As the case for phytoplankton, zooplankton, fish, and invertebrate larvae transported by the filament towards the islands, mesopelagic fish were suggested to drift in the filament structure during their residence time at night in shallower layers because of their diel vertical migration. In this way, these organisms could be transported in a rather food-rich environment (zooplankton and other larvae in the filament) towards the anticyclonic eddy south of the island and then to over the island shelf by stronger currents. We posited that the strandings that occurred during dawn over the narrow shelf of the island were related to the feeding activity of these pelagic fish.

Interestingly [30,57], in the Mediterranean Sea, it was also concluded that the new and full moon influenced the strength of the currents. Days before to the stranding, we observed a full moon (26 May 2021, Figure 5A,C), which coincided with an increase in both the current and wind velocity during this period. However, during the stranding day, we recorded that the moon was in the last quarter. During the full moon, the vertical migration of zooplankton and micronekton to the upper layers of the water column is less than at other times [87,88,89,90,91], but the during the new and full moon phase the highest tidal intensity occurs [92]. Therefore, the migration occurred in the moon’s last quarter phase, and when the fish had already migrated upwards, this stronger tidal current, in combination with the upwelling transport and eddy circulation, could have facilitated the transport of the fishes deep inside the island shelf region.

## 5. Conclusions

The stranding of mesopelagic fish recorded in the south of Gran Canaria Island was the first documented evidence of this process in the Canary Islands and the Macaronesian region (Madeira, Azores, and Cape Verde Islands). Considering the volcanic nature of the islands (with narrow shelves) and the year-round presence of mesoscale oceanographic structures such as island wakes, eddies, and upwelling filaments, we suggest that this kind of stranding event should be common. In the future, the implementation of a monitoring program of these coastal sites in combination with satellite-derived environmental data will help to gather robust information about such events and what species are associated with these processes. This data will also allow us to assess changes in species richness.

## Figures and Tables

**Figure 1 animals-12-03465-f001:**
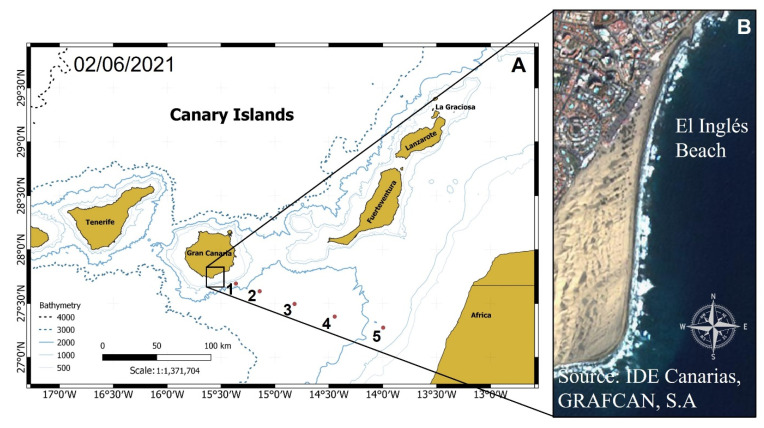
Location of stranded mesopelagic fishes along “El Inglés” Beach (Gran Canaria Island, Spain). (**A**) Map of the eastern islands of the Canarian Archipelago. Numbers on the map are the locations where oceanographic variables were examined. (**B**) Source: IDECanarias, GRAFCAN, S.A. Canary Islands Government (2021).

**Figure 2 animals-12-03465-f002:**
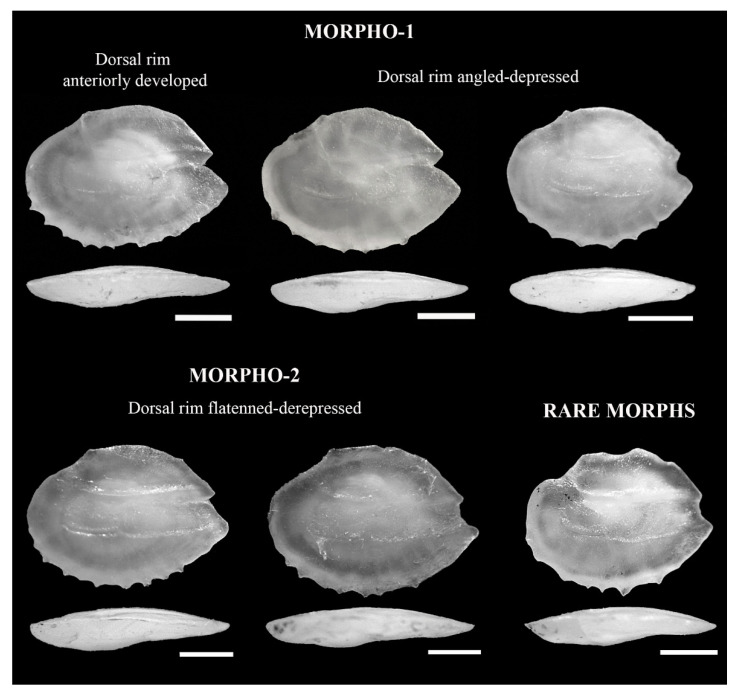
Image composition of otolith morphotypes according to the variability of shapes obtained from mesopelagic fish along the southeast coast of Gran Canaria island.

**Figure 3 animals-12-03465-f003:**
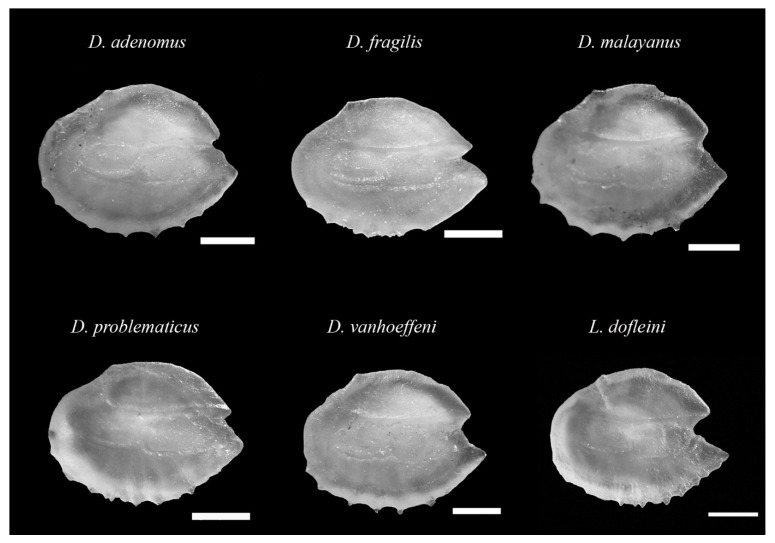
Morphology of otoliths extracted in the mesopelagic fish species collected along the southeast coast of Gran Canaria.

**Figure 4 animals-12-03465-f004:**
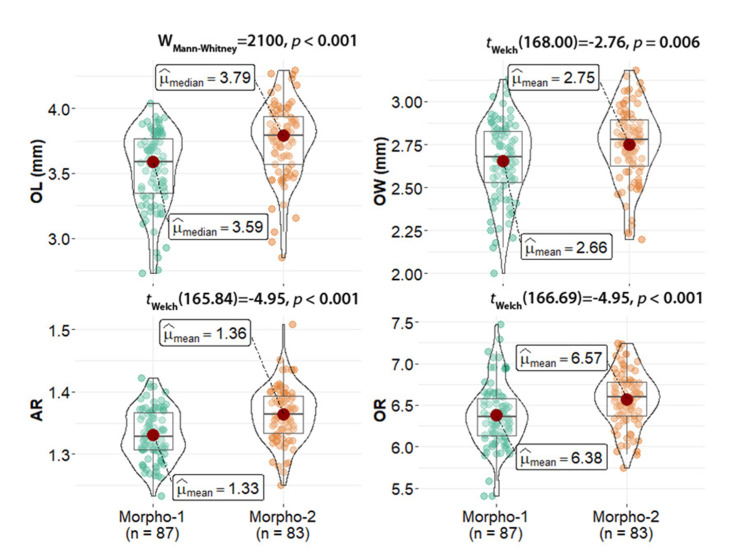
Otolith length (OL), width (OW), aspect ratio (AR), and otolith relative length (OR) for each specimen analyzed.

**Figure 5 animals-12-03465-f005:**
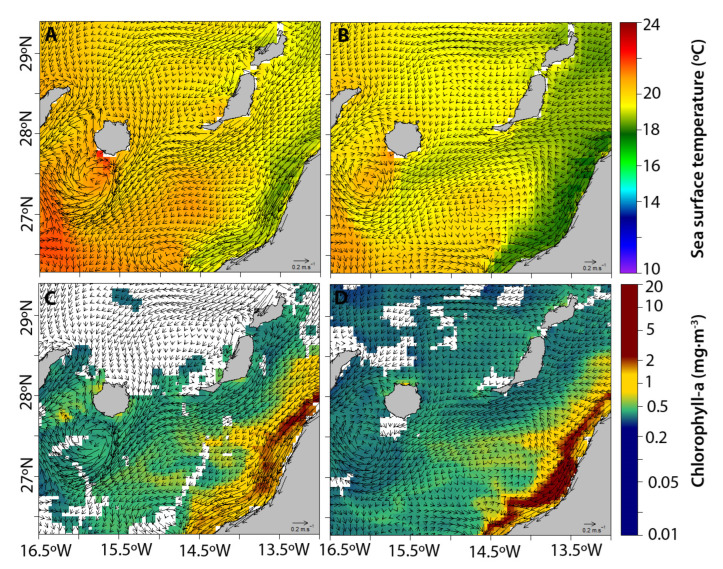
(**A**,**B**) Remotely sensed image of sea surface temperature and (**C**,**D**) chlorophyll a concentration between Gran Canaria Island and the coastal zone off northwest Africa during 26th of May (**A**,**C**) and 2 June (**B**,**D**), 2021. Data was downloaded from the Copernicus Marine platform and NASA’s OceanColorWeb. Current direction and velocity are represented by black arrows. Velocity is proportional to the length of the arrows.

**Figure 6 animals-12-03465-f006:**
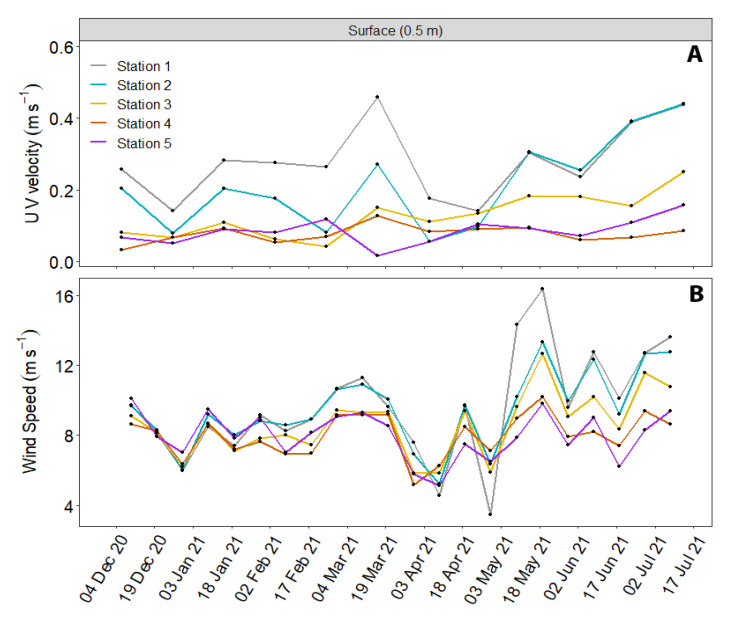
(**A**) Total component of current velocity (m·s^−1^) and (**B**) wind speed (m·s^−1^) from December 2020 to July 2021. Data were download through the Copernicus Marine platform. The velocity of the current due to the filament and the anticyclonic eddy was higher at Stations 1 and 2 near Gran Canaria Island during the stranding event (2 June 2021).

## Data Availability

The raw data supporting the conclusions of this article are publicly available through AFORO website (http://aforo.cmima.csic.es/ (accessed on 12 August 2022)) and PANGEA platform: Sarmiento-Lezcano, Airam Nauzet; Couret, María; Lombarte, Antoni; Olivar, M Pilar; Landeira, José María; Hernández-León, Santiago; Tuset, Victor M (2022): Otolith morphological measures of stranding mesopelagic fishes in the Canary Islands during June 2021. PANGAEA, https://doi.org/10.1594/PANGAEA.951480 (accessed on 12 August 2022).

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
