# Peer review of "Stranding of Mesopelagic Fishes in the Canary Islands"

_animals, 2022, doi:10.3390/ani12243465_

Round 1

Reviewer 1 Report

Dear Editor

The manuscript fits within the scope of the journal Animal. The authors well described the event of stranding around the Canary Islands. Namely, they comprised all available data that included the determination of stranded fish as well as all abiotic and/or biotic parameters. Considering all of that they successfully managed to explain the occurrence of this event. I think that in this form this manuscript can be classified as a short Communication paper that can be accepted for publication in this respected journal.

Nevertheless, I think that the authors put a lot of effort to obtain this data set hence, I would like to suggest them to try free software ICHTHYOP (https://ichthyop.org/) which can incorporate all the data they have and it will produce the pattern of this species transport and finally their stranding.

Author Response

The reviewer #1 suggestion is a very good idea. However, I have tried to use the software ICHTHYOP and I have not been able to create outpouts with my own data and the software examples. Maybe, I could use it in the future.

Reviewer 2 Report

First, nice paper!  Second, I have returned to the Editors a marked up copy of the PDF.   My purpose was to improve the readability of the paper.  As I noted at the top of the MS, I tried to removed the word "the" whenever possible.  I also recommend that you write in the first person -- activate the sentences.  

If you have questions about my comments, as the editor to forward them to me.  -- R.L. Wallace

Author Response

According the comments in the manuscript:

Point 1: In line 137 (now line 140). Some what unclear.

Response 1: We have modifed the text as follows: “than antirostrum showing variable shape”.

Point 2: In line 156 (now in line 160). Unclear what is developed and depressed.

Response 2: We have modifed the text as follows: “and mopho-type 2 with dorsal predorsal rims lesser developed and depressed in comparison to morphotype-1 (Figure2).

Point 3: In line 178-180 (now in line 182). Unclear.

Response 3: Agreed. We suggest to modify the text how “ The total component of current velocity and wind values were higher near to the island (Station 1 and 2) than in open ocean as the effect of the upwelling filament and the anticyclonic eddy (Figure 5B, D) during the previous month.”

Point 4: In line 251 (now in line 254-255). Unclear, did I help?

Response 4: Author do not understand what the reviewer does not understand in this case. However, we will to try explain de sentence. Most of the mesopelagic fishes perform vertical migration from mesopelagic zone to surface during the night. Taking into account, and that the phytoplankton, zooplankton and ichthyoplankton are transported by the filament towards the islands, we suggest that the mesopelagic fishes are also transported by the upwellig fillament when they performed the vertical migration at night.
